# Genome-Wide Identification of *GATA* Family Genes in *Phoebe bournei* and Their Transcriptional Analysis under Abiotic Stresses

**DOI:** 10.3390/ijms241210342

**Published:** 2023-06-19

**Authors:** Ziyuan Yin, Wenhai Liao, Jingshu Li, Jinxi Pan, Sijia Yang, Shipin Chen, Shijiang Cao

**Affiliations:** 1College of Forestry, Fujian Agriculture and Forestry University, Fuzhou 350002, China; y596323048@163.com (Z.Y.); lwh1623850793@163.com (W.L.); chuchu7613@163.com (J.L.); 17726282721@163.com (J.P.); 18816401621@163.com (S.Y.); fjcsp@126.com (S.C.); 2University Key Laboratory of Forest Stress Physiology, Ecology and Molecular Biology of Fujian Province, College of Forestry, Fujian Agriculture and Forestry University, Fuzhou 350002, China

**Keywords:** *GATA* family genes, *Phoebe bournei*, expression patterns, evolution, abiotic stress

## Abstract

GATA transcription factors are crucial proteins in regulating transcription and are characterized by a type-IV zinc finger DNA-binding domain. They play a significant role in the growth and development of plants. While the *GATA* family gene has been identified in several plant species, it has not yet been reported in *Phoebe bournei*. In this study, 22 *GATA* family genes were identified from the *P. bournei* genome, and their physicochemical properties, chromosomal distribution, subcellular localization, phylogenetic tree, conserved motif, gene structure, cis-regulatory elements in promoters, and expression in plant tissues were analyzed. Phylogenetic analysis showed that the *PbGATAs* were clearly divided into four subfamilies. They are unequally distributed across 11 out of 12 chromosomes, except chromosome 9. Promoter cis-elements are mostly involved in environmental stress and hormonal regulation. Further studies showed that *PbGATA11* was localized to chloroplasts and expressed in five tissues, including the root bark, root xylem, stem bark, stem xylem, and leaf, which means that *PbGATA11* may have a potential role in the regulation of chlorophyll synthesis. Finally, the expression profiles of four representative genes, *PbGATA5*, *PbGATA12*, *PbGATA16*, and *PbGATA22*, under drought, salinity, and temperature stress, were detected by qRT-PCR. The results showed that *PbGATA5*, *PbGATA22,* and *PbGATA16* were significantly expressed under drought stress. *PbGATA12* and *PbGATA22* were significantly expressed after 8 h of low-temperature stress at 10 °C. This study concludes that the growth and development of the *PbGATA* family gene in *P. bournei* in coping with adversity stress are crucial. This study provides new ideas for studying the evolution of GATAs, provides useful information for future functional analysis of *PbGATA* genes, and helps better understand the abiotic stress response of *P. bournei*.

## 1. Introduction

The growth and development of plants is a continuous process typically starting from seed germination and ending with seed maturation. During these stages, plants must face and respond to a variety of environmental conditions. Plant responses to environmental challenges are commonly mediated through transcription factors that regulate gene expression of their target genes via cis-acting elements in the promoter. Therefore, the study of transcription factors is important to understand genetic control of gene expression in many plant metabolic pathways [1,2]. Many well-known families of transcription factors have been demonstrated in plants, such as GATA (GATA-binding factor) [3], bZIP (Basic leucine zipper) [4], MYB (Myeloblastosis) [5], NAC [6,7,8,9,10], bHLH (Basic helix–loop–helix) [11], ERF (Ethylene response factor) [12], CBF (CRT-binding factor) [13], and WRKY [11].

GATA-binding transcription factors are a family of proteins found in eukaryotes, including fungi, animals, and plants [14,15,16]. These proteins contain one or two highly conserved zinc finger DNA-binding domains that match the common sequence C-X_2_-C-X_18–20_-C-X_2_-C, followed by a basic region [17,18]. Fungal GATA factors are known to regulate various processes such as nitrogen metabolism, photo-induction, siderophore biosynthesis, and mating-type switching [14]. These factors typically contain the C-X_2_-C-X_17_-C-X_2_-C or C-X_2_-C-X_18_-C-X_2_-C domains [14]. Six *GATA* family members have been found in vertebrates, and there is a certain evolutionary relationship between them [15]. The GATA family of transcription factors is involved in multiple processes of plant growth and development and plays an important role in biological processes such as abiotic stress and secondary metabolism [16].

GATA factor was first discovered in tobacco (*Nicotiana tabacum*) and was found to be similar to the *NIT2* gene in *Neurospora crassa*. The *Ntl1* gene, which encodes the GATA factor, is believed to regulate the nitrate assimilation pathway due to its weak nitrate inducibility and regulation by light [19]. Subsequently, the GATA family has been identified in several plant species, including *Arabidopsis thaliana* [16], *Oryza sativa* [16,20], *Eucalyptus urophylla* [3], and *Solanum lycopersicum* [21]. GATA factor has been shown to inhibit the flowering of *A. thaliana* [22]. Research has demonstrated that LLM-domain B-class *GATA* genes are involved in the control of stomata formation, and these genes operate before the regulators of stomata formation, namely *SPEECHLESS* (*SPCH*), *MUTE*, and *SCREAM/SCREAM2*, and also function in conjunction with or separately from the patterning regulators *TOO MANY MOUTHS* and *STOMATAL DENSITY AND DISTRIBUTION1* [23]. In the previous study, seven cucumber *GATA* genes were involved in chloroplast development and chlorophyll biosynthesis, four cucumber *GATA* genes were also associated with low nitrogen, and there were six cucumber *GATA* genes with both anti-abiotic and biotic stress functions [24]. In Chinese pears (*Pyrus bretschneideri*), the *GATA* gene has been found to play an important role in hormone signaling pathways [25]. In addition, *GNC* and *CGA1* have been shown to play an important role in plant nitrogen assimilation [26]. Previous studies have indicated that GATA transcription factors play a significant role in various physiological and biochemical processes, including but not limited to plant photoresponse, chloroplast development, chlorophyll biosynthesis, low nitrogen response, hormone synthesis, and regulation of plant response to drought and salinity stress. Despite the vast size of the *GATA* gene family, the majority of *GATA* genes’ biological functions remain unexplored and require further investigation.

The diversity of forest ecosystems has brought great benefits and impacts to people’s life, while forests also regulate the carbon cycle in the atmosphere and alleviate problems such as soil acidification [27,28,29]. *Phoebe bournei* is widely distributed in southern China and is one of the important tree species that make up forests. *Phoebe bournei* is evergreen trees or shrubs of the genus Phoebe in the *Lauraceae* family and is a globally endangered species in the *Lauraceae* family [30]. It has a wood aroma, and its tough material, which is not easy to crack and process, is widely used in wood carving art and building construction, with high economic value and ecological value. However, extreme environments such as cold stress and drought stress hinder the growth of *P. bournei* [31,32]. Transcription factors have a profound effect on enhancing plant cold and salinity tolerance [33,34,35,36,37], helping plants cope with extreme temperatures [38,39,40,41], and regulating plant hormones [42,43]. GATA transcription factors, in particular, have been found in numerous plants but have not been explored in *P. bournei*. Therefore, this study systematically analyzed the GATA transcription factor of *P. bournei* and predicted and analyzed its physicochemical properties, phylogenetic relationships, chromosome localization, gene structure, protein-conserved group sequence, cis-acting element, and qRT-PCR detection, which provided a solid foundation for further improving the stress resistance of *P. bournei* and exploring the *GATA* family gene.

## 2. Results

### 2.1. Identification of PbGATA Genes in P. bournei

A total of 22 *GATA* genes were identified in the *P. bournei* genome, and these genes were renamed as *PbGATA1~PbGATA22* according to their distribution positions on the *P. bournei* chromosome (Table 1 and Figure 1). The 22 GATA proteins encode amino acids between 139 aa and 835 aa. The size of a protein is usually proportional to the length of its amino acid sequence. In this study, the 22 PbGATA proteins had a relative molecular weight that ranged from 15,586.05 Da (PbGATA18) to 94,357.56 Da (PbGATA11). In total, 12 PbGATA proteins were acidic (pI < 7.0), and the remaining 10 were alkaline. Moreover, they were all unstable proteins (instability index > 40), with a range of 41.87 to 76.26. The results demonstrated that the aliphatic index ranges from 50.73 to 84.42, with an average value of 62.82, reflecting the thermal stability of the PbGATA protein. The PbGATA proteins were all hydrophilic proteins, according to the grand average of hydropathicity, which was negative. Approximately 16 PbGATA proteins were predicted to be localized in the nucleus via subcellular localization, followed by five in the cytoplasm, and one in the chloroplast, named PbGATA11.

### 2.2. Phylogenetic Analysis and Sequence Alignment of GATA Proteins

A maximum likelihood phylogenetic tree was created using MEGA 7.0.21 software based on the multiple sequence alignment of 22 PbGATA proteins, 30 AtGATA proteins, and 35 MdGATA proteins in order to examine the phylogenetic relationship and biological function of the *GATA* genes among various species and classify the *GATA* genes discovered in *P. bournei* (Figure 2). The PbGATA family proteins were classified into four clusters (A, B, C, and D) based on the classification of *Arabidopsis* and *M*. *domestica* GATA proteins [18,44]. Subfamily A had the most PbGATA proteins (nine), accounting for 40.9% of the total PbGATA proteins among the four categorized subfamilies. This was followed by subfamilies B (seven), C (five), and D, the last of which has the fewest GATA proteins with only one member (4.54%), PdGATA5. It is thought that PbGATA retains all four subfamilies across evolution, but the number of *GATA* genes varies between species.

Their conserved domain sequences were aligned in order to further investigate the 22 PbGATA proteins’ sequence characteristics. The multiple sequence alignment showed that the conserved domain C-X_2_-C-X_18–20_-C-X_2_-C was present in all GATA proteins (Figure 3 and Figure 4B), and its secondary structure includes four β collapse and an α helix. The analysis showed that the zinc finger domains of most GATA amino acid sites in *P. bournei* were conservative.

### 2.3. PbGATA Protein Gene Structure and Conserved Motif Analysis

Exon/intron organization analysis of 22 *PbGATA* genes indicated that the number of exons in *GATA* genes ranged from 1 (*PbGATA20*) to 18 (*PbGATA11*). Subfamily B had the fewest average exons per gene (three), whereas subfamily C had the most (nine). Except for subfamily A, the structural properties of the *PbGATA* genes within the same subfamily were comparable but differed between subfamilies (Figure 4). For instance, each *PbGATA* gene in subfamily C had eight or more exons, whereas the genes in subfamily B had two or three exons, with a maximum of five. With the exception of *PbGATA11*, all components of subfamily A have comparable gene structure features.

The 22 PbGATA proteins had a total of 10 conserved motifs, denoted as motifs 1 through 10. Similar conserved motif compositions were frequently present in the majority of the GATA proteins from the same subfamily (Figure 5B). In total, five of the 22 PbGATAs only contained motif 1. Overall, nine of the 22 PbGATAs contained motif 1 and 2. In addition, nine of the 22 PbGATAs contained motifs 2 and 1, and five contained motifs 1 and 10. Motif 1 occurs in all subfamilies, and, according to Figure 3 and Figure 5C, this motif is annotated as the GATA zinc finger domain. Individual motifs occur only in specific subfamilies, for example, motifs 5 and 9 occur only in subfamily A, motif 8 occurs only in subfamily B, and motifs 6, 7, 3, 4, and 10 occur only in subfamily D.

As a consequence, the *PbGATA* subfamily categorization is further supported by shared gene structures, conserved motif arrangements, and phylogenetic trees within the same subfamily. The variety of the quantity, arrangement, and distribution of diverse motifs in different subfamilies may be what differentiates them from one another.

### 2.4. Cis-Acting Elements Analysis of the PbGATA Gene Family

By detecting 2000 bp promoter sequences upstream of the *PbGATA* gene, studies show 24 cis-acting elements in the promoter region of the *PbGATA* family, such as ARE, AAGAA-motif, ABRE, Box 4, and so on, which involve four components of environmental stress reaction: plant growth and development, hormone response, and light response (Figure 6 and Appendix A). Among them, the largest number of elements of function is the classification of stress response (285), followed by the classification of hormone response (262), and the smallest is the classification of plant growth and development (111). The drought related cis-element MYB (91), the pressure-resistant cis-element STRE (79), and the MeJA-responsiveness-related element MYC (123) were present in 22 *PbGATAs,* in large quantities. The results suggest that *PbGATA* family genes may be more sensitive to stress and hormonal responses.

The subfamilies A, B, C, and D contained 353, 271, 139, and 47 cis-acting elements, respectively, of which the subfamily A *PbGATA19* (50) contained the largest number of cis-elements, followed by *PbGATA17* (49) of subfamily B and *PbGATA5* (47) of subfamily D; CAT-boxes were the least absent (7). The WUN-motif is not present in subfamily C, which may be due to the weak response of the subfamily to wounds; subfamily B does not have the CCGTCC-box cis-acting element, possibly because this subfamily plays a less important role in plant growth. In a nutshell, the cis-elements study suggested that while a considerable number of *PbGATA* genes are expected to respond to diverse environmental stresses, their impact on plant growth and development is rather minor.

### 2.5. The Distribution, Genomic Synteny, and Gene Duplication of PbGATA Genes

According to the *P. bournei* genome annotation, 22 *PbGATA* genes were mapped to eleven chromosomes, except chromosome 09 (Figure 1). Of these, chromosome 05 had the most *GATA* genes (up to five), followed by chromosome 02, which had four *GATA* genes. Tandem replication events are defined as two or more closely related genes scattered in the range of 200 KB, whereas fragment replication events are defined as pairs of homologous genes placed on distinct chromosomes [45]. In the *PbGATA* genes, we found no tandem repeats (Figure 1). In addition, fragment repeat events were identified using TBtools (Figure 7). *PbGATA1/13*, *PbGATA5/12*, *PbGATA6/14*, *PbGATA6/16*, *PbGATA11/21,* and *PbGATA14/16* were among the genes with a fragment replication event. Of the six pairs of genes with fragment repeat events, three pairs of *PbGATA* genes belonged to subfamily C, two pairs belonged to subfamily A, and one pair belonged to subfamily B.

The results showed that these duplication episodes are the major driving factor behind the increase in the number of *PbGATA* genes, and subfamily A and subfamily C, which contain a relatively significant number of *PbGATA* genes, may have been enlarged throughout the entire genome duplication process. Segmental duplication events may be important in the increase in the number of *PbGATA* genes in *P. bournei*.

To further study the evolutionary mechanism of the *PbGATA* family, the collinearity of *PbGATA* gene pairs between the *Arabidopsis* genome, the *Oryza sativa* genome, the *Populus thichocarpa* genome, and the *Glycine max* genome was compared (Figure 8). The results showed that *PbGATA* formed 13 collinearity gene pairs with *AtGATA*, 17 collinearity gene pairs with *OsGATA*, 26 collinearity gene pairs with *PtGATA*, and 36 collinearity gene pairs with *GmGATA*.

Multiple *PbGATA* genes have been identified as homologous genes of a single *AtGATA*, *OsGATA*, *PtGATA*, and *GmGATA* gene. Similarly, there are multiple *AtGATA*, *OsGATA*, *PtGATA*, and *GmGATA* genes that are homogeneous to a single *PbGATA* gene. From this collinearity relationship, we can think that the amplification of this gene family may occur before the differentiation of *P. bournei*, *Arabidopsis*, rice, etc.

### 2.6. Expression Analysis of PbGATAs in P. bournei Tissues

The expression patterns of the 22 *PbGATAs* were compared in five *P. bournei* tissues: the root bark, the root xylem, the stem bark, the stem xylem, and the leaf (Figure 9 and Appendix A). The lowest expression was in the root xylem and the greatest was in the stem bark in five separate tissues, and the overall expression of bark tissue was higher than the total expression of the xylem and leaves. In subfamily A, except for *PbGATA19*, which was only expressed in the bark and leaf tissues, the remaining eight genes were expressed in various tissues, of which the expression of *PbGATA11* was the highest among the *22 PbGATAs*. In subfamily B, *PbGATA18*, which was expressed in all five tissues, and the remaining genes were only expressed in some tissues. For example, *PbGATA10* was only expressed in the root bark and was the least expressed. *PbGATA12* and *PbGATA3* were expressed at low levels or absent in other tissues but were at high levels in the leaf tissue, presumably due to the fact that these genes are largely involved in leaf growth and development. Subfamily C genes were expressed in all five tissues and had the highest average expression. While subfamily D has just one gene, it was expressed in all five organs. These findings suggested that the expression levels of *PbGATA* vary within and between the subfamilies.

### 2.7. Expression of PbGATA Genes under Abiotic Stress

Four representative genes, *PbGATA5*, *PbGATA12*, *PbGATA16*, and *PbGATA22*, were chosen in order to study their expression in response to temperature, salinity, and drought stress. These four genes originated from four subfamilies, all of which had more stress-related cis-acting elements and were highly expressed in the leaves (Figure 5 and Figure 9). The results showed that drought, salt, and temperature affected the expression levels of the *PbGATA* gene (Appendix A). Under the immersion treatment of 10% PEG6000 nutrient solution (Figure 10), the expression of *PbGATA5* and *PbGATA22* increased significantly by about 10 times 8 h after treatment compared with before the start of treatment, and the expression of *PbGATA16* was extremely high. Based on these results, we suspect that *PbGATA5*, *PbGATA22*, and *PbGATA16* play an important role in coping with drought stress. Under 10% NaCl stress, we found that *PbGATA5* and *PbGATA12* were inhibited to varying degrees, but the upregulation of *PbGATA16* and *PbTAGA22* began at 4 h after treatment, and *PbTAGA22* had higher upregulation than *PbGATA16*.

Under 40 °C treatment, the expression of each gene showed varying degrees of downregulation, indicating that the expression of the *PbGATA* gene was not visible under high-temperature stress. Each gene changed in distinct ways when exposed to 10 °C compared to the control group. The expression of *PbGATA12* and *PbGATA22* was particularly high after 8 h of treatment. Therefore, we hypothesize that *PbGATA12* and *PbGATA22* have a significant effect on low-temperature stress in *P. bournei*.

## 3. Discussion

GATA transcription factors have been demonstrated to play an important role in plant salt tolerance [21,46], temperature stress [47], chlorophyll production [26], and hormone treatment [48]. Because of the importance of the *GATA* gene family in abiotic stress and plant growth and development, it has been studied in plants such as peanuts [49] and cucumbers [24], but genome-wide discovery of the *PbGATA* family gene has not been pursued. As a result, genome-wide characterization and expression investigation of the *GATA* gene family will aid in our understanding of the *GATA* gene’s function and involvement in *P. bournei*.

In this research, we used bioinformatics to identify the *PbGATA* family gene and discovered 22 *PbGATA* genes separated into four subfamilies (Figure 2). Like *A. thaliana* and *M. domestica* [18,44], *PbGATA* is distributed in the highest number in subfamily A, subfamily B is second, and in subfamily D, it is the least numerous. We found that *P. bournei* had eight and 13 GATA members less than *Arabidopsis* and apple, respectively. There are different evolutionary branches of the PbGATA family in *P. bournei*. The GATA family members of the three were cross-distributed, and no separate branches were found (Figure 2). This proves that the *GATA* family gene of dicots has no obvious variation in the evolutionary direction for the time being and is still conserved.

The leaf-transcriptome profiles of *P. bournei* treated with varying drought stress revealed that numerous domains, including the GATA zinc finger domain, displayed distinct expression patterns between the treatment’s internal conjunction processes [32,50,51]. The GATA paralog transcription factors *GNC* and *CGA1* regulated nitrogen assimilation in green tissues by modulating the expression of chloroplast-localized *GLUTAMATE SYNTHASE* (*GLU1*/*Fd-GOGAT*), hence influencing the number of chloroplasts and the proportionate transcription level of leaf starch [26,52]. *GATA* genes *Csa3G165640*, *Csa5G622830*, *Csa3G843820*, *Csa6G405920*, *Csa6G502700*, *Csa6G504690*, and *Csa7G452960* have also been implicated in chlorophyll biosynthesis regulation in cucumbers [24]. The *GATA* genes *GmGATA58* and *PdGATA19* have been shown to be important in controlling chlorophyll production in soybean [53]. Several of the preceding investigations revealed that various *GATA* genes are directly connected to the physiological function of chlorophyll, influencing plant physiological processes. It is worth exploring the fact that our analysis of the predictions of *PbGATA* subcellular localization shows that most *PbGATA* is localized in the cytoplasm and nucleus, and only *PbGATA11* is localized in the chloroplasts (Table 1). Moreover, *PbGATA11* has the largest molecular weight, and the expression is also highest in the root epidermis, root xylem, stem epidermis, stem xylem, and leaves (Figure 9). We speculate that the function and role of this gene are related to the physiological activity of chloroplasts, which needs to be proven by further experiments.

Variations in conserved motif and gene structure across GATA protein family members are key factors for functional variability [49,54]. A PbGATA motif study revealed that ZnF_GATA, a critical domain for identifying *PbGATA* genes, was strongly conserved in virtually all *PbGATA* genes (Figure 4). The *PbGATA* gene’s conserved domain was discovered by multiple sequence alignment to be C-X_2_-C-X_18–20_-C-X_2_-C, which is compatible with the conserved domain previously identified in peppers [48], whereas two subfamilies of cabbage rape have the N-X_2_-C-X_18_-CX_2_C domain, not the C-X_2_-C-X_18–20_-C-X_2_-C conserved domain, and one subfamily of the *GATA* gene of cucumbers has the C-X_4_-C-X_18_-C-X_2_-C domain [24,55]. Baseline analysis and replication events established links between subfamily A and the remaining three subfamilies of the *PbGATA* family gene [56]. There is a correlation of amino acid sequences between subfamily A and subfamilies B, C, and D of the *GATA* family gene [57]. Only *St4CL1* in subfamily A in the potato 4CL family gene has motif 6, and all St4CLs in subfamily B contain motif 6, and an important reason for its phylogenetic differentiation is the change in amino acids in motif 6 [58]. In the conservative motifs analysis of this study, motif 9 is only distributed in subfamily A. This suggests that motif 9 may be the evolutionary cause of subfamily A of the *PbGATA* gene (Figure 5). *St4CL1* and *St4CL2* are tandem replicate genes in potato subfamily A; however, *St4CL2* lacks motif 6, indicating that one of the reasons the *St4CL* family gene split into two forms was due to the loss of motif 6 in this gene tandem repeat event [58]. In the fragment replication event of *PbGATA*, *PbGATA5* is a fragment replication gene with *PbGATA12*, but *PbGATA12* does not have motif 3, motif 4, and motif 10, and motif 8 is added, which indicates that the deletion of motif 3, 4, and 10 and the increase in motif 8 occur in this gene fragment duplication event, which may lead to the differentiation of the *PbGATA* family gene into subfamily B and subfamily D types from here. Thus, motifs 3, 4, 10, and 8 are also genetic and phylogenetic connections between subfamilies B and D.

The response of genes to stress Is closely related to the cis-element present in the gene [59]. For example, CBF1 can promote C-repeat/DRE binding to cope with low-temperature and dehydration stress in *Arabidopsis* [60]. The *CaGATA* gene in chickpeas regulates the response to water stress [61]. The qRT-PCR analysis showed that the *PbGATA* gene had a relatively obvious response to PEG6000-simulated drought, salt, and 10 °C low-temperature stress (Figure 10). At the same time, cis-acting elements linked with drought stress, such as MYB (drought-related element), were found in the promoter region of the *PbGATA* genes, including *PbGATA16*, *PbGATA5*, and *PbGATA22* (Appendix A). Among the four representative genes selected, *PbGATA16* had the most MYB cis-elements. Therefore, it is reasonable to guess that *PbGATA5*, *PbGATA22*, and especially *PbGATA16* play an important role in the response to drought stress in *P. bournei*. The upstream signal of the ICE-CBF cascade is JA, and cold stress induces the accumulation of endogenous JA, actively modulating the cold tolerance of *A. thaliana*. Plant hormones have an important influence on regulating plant metabolism, salt resistance, and drought tolerance [62]. In *OsLPXC* knockout plant leaves in rice plants under low-temperature stress, a large accumulation of MDA and electrolyte leakage occur with the inhibition of JA biosynthesis [63]. In this study, the four representative genes were upregulated to varying degrees under low temperature treatment at 10 °C. We found that MYC (MeJA-responsiveness) promoters are present in the cis-component of these genes and in the largest number in the *PbGATA12* gene. Cold promotes the accumulation of JA, leading to MYC2 regulating *ADC1* expression, which in turn prompts tomato plants to cope with low-temperature stress [64,65]. Therefore, we suspect that the function of *PbGATA* in response to low-temperature stress is related to the MeJA response regulated by the cis-element. Based on the above findings, the role of *PbGATA* genes under drought and low-temperature stress in *P. bournei* can be predicted (Figure 11) [64,66].

## 4. Materials and Methods

### 4.1. Identification of PbGATA Genes in P. bournei

The *P. bournei* genome assembly file was downloaded from China National GeneBank DataBase [67] (CNGBdb) (https://db.cngb.org/ (accessed on 5 December 2022)). A specific hidden Markov model (HMM) of GATA transcription factor [68] (Pfam number: PF00320) was used to search for the candidate *GATA* genes in the *P. bournei* genome. Then, we further used the NCBI Conserved Domain Database (NCBI-CDD) (https://www.ncbi.nlm.nih.gov/Structure/bwrpsb/bwrpsb.cgi (accessed on 5 December 2022)) to verify the candidate sequences and deleted genes that did not belong to the GATA transcription factor family [69]. Ultimately, a total of 22 *PbGATA* genes were identified and renamed according to their position on the chromosomes. Furthermore, the ExPASy online tool (http://www.expasy.ch/tools/pi_tool.html (accessed on 5 December 2022)) was used to determine the physicochemical characteristics of all PbGATA proteins, including amino acid number (size), molecular weight (MW), theoretical isoelectric point (pI), instability index, aliphatic index, and grand average of hydropathicity (GRAVY). Finally, we predicted the subcellular localization of PbGATA proteins using WoLF PSORT (https://wolfpsort.hgc.jp/ (accessed on 6 December 2022))

### 4.2. Phylogenetic Analysis

GATA protein full-length sequences of *Arabidopsis* and *M. domestica* were obtained from PlantTFDB (http://planttfdb.gao-lab.org/ (accessed on 6 December 2022)) [70]. Multiple sequence alignment of three plants’ GATA protein amino acids was conducted by MUSCLE function in MEGA 7.0.21 with default parameters [71]. Additionally, the sequence alignment results were further visualized with Jalview 2.11.2.6 software [72]. The MEGA 7.0.21 program was applied to produce the phylogenetic tree using the maximum likelihood (ML) approach with the best-fit model “JTT + F + I” and 1000 bootstrap replication times. The phylogenetic trees were visualized and further enhanced with the iTOL website (https://itol.embl.de/itol.cgi (accessed on 6 December 2022)).

### 4.3. Gene Structures, Conserved Domain and Protein Motifs Analysis

Distribution information on exons and introns was obtained from *P. bournei* genome GFF files [67]. The conserved domain of the PbGATA proteins was uploaded and verified by the NCBI-CDD Database [69]. Conservative motifs of protein sequences were exhibited by MEME suite (http://meme-suite.org/tools/meme (accessed on 6 December 2022)) [73] with the following two parameters: the maximum motif number was 10 and the distribution of motif site occurrences was zero or one per sequence. Finally, the intron–exon structure, conserved domain, and 10 motifs of the PbGATA proteins were illustrated by Tbtools [74].

### 4.4. Chromosomal Location, Gene Duplication, and Collinearity Relationship

Information on the location of the *PbGATA* genes was found based on the GFF annotation files of the *P. bournei* genome [67]. Taking advantage of MG2C v2.1 (http://mg2c.iask.in/mg2c_v2.1/ (accessed on 7 December 2022)), the chromosome location pattern was produced. Gene duplication patterns of the *PbGATA* family gene were identified and examined by Tbtools [74]. The genome files of four other plant species, namely, *Arabidopsis*, rice, poplar, and soybean, were obtained from the NCBI. Then, the collinearity relationships of *PbGATA* with the four above species were analyzed and plotted using the “Advanced Circos” functional plate in Tbtools [74].

### 4.5. Cis-Elements in the Promoter and Expression Analysis of PbGATA Genes

The 2000 bp upstream sequence of *PbGATAs* were extracted and served as the promoter sequence used to identify cis-elements and prediction using PlantCARE (https://bioinformatics.psb.ugent.be/webtools/plantcare/html/ (accessed on 7 December 2022)) [75]. The positions and numbers of the cis-elements were visualized by Tbtools [74]. RNA-seq data of different tissues in *P. bournei* were downloaded from the NCBI database using BioProject accession number PRJNA628065 [69].

### 4.6. Plant Materials and Abiotic Stresses Treatment

One-year-old *P. bournei* seedings were collected from Fujian Academy of Forestry and grown under natural conditions. During the stress treatment period, the seedlings were cultured in an artificial climate incubator with a temperature of 25 °C and a humidity of 75%. Then, the *P. bournei* seedlings were exposed to salt stress (10% NaCl solution), drought stress (10% PEG6000), cold stress (4 °C), and heat stress (40 °C). Samples of mature leaves were collected at 0 (CK), 4, 6, 8, 12, and 24 h. The control group (CK) was treated with distilled water and/or normal growth conditions. Then, the *P. bournei* leaves obtained after treatment were immediately placed in liquid nitrogen and stored in a refrigerator at −80 °C for further RNA extraction.

### 4.7. RNA Extraction and qRT-PCR Analysis

Total RNA of the collected leaf tissue was extracted from the control group and the stress-treated samples using a HiPure Plant RNA Mini Kit (Magen). cDNA was synthesized with a PrimeScript RT reagent Kit (Perfect Real Time) (TaKaRa). qRT-PCR was conducted to evaluate the expression profiles of *PbGATA* genes in response to stress treatment. The specific primers used in the qRT-PCR experiment were designed by the Primer 3 website (http://bioinfo.ut.ee/primer3-0.4.0/ (accessed on 27 March 2023)) and are listed in Appendix A. *PbEF1α* was obtained as a reference gene (GenBank number, KX682032.1) [31]. The raw Cq values were analyzed by the 2^−ΔΔCT^ method and compared with the reference gene [76]. All experiments were performed with three biological replicates and three technical replicates. The relative transcriptional expression levels were subjected to one-way ANOVA and multiple comparisons with the control group at the 5% significance level via GraphPad Prism 8.3.0 software.

## 5. Conclusions

In conclusion, the PbGATA genes are crucial for the growth of *P. bournei* and hence have vital potential for the enhancement of this extremely commercially relevant woody plant. These genes were classified into four subfamilies based on a phylogenetic study and their gene structure, which is congruent with the previously stated GATA family. Our findings shine a spotlight on the evolution of the *GATA* family gene in angiosperms. PbGATA genes are hydrophilic proteins that are unstable. The conserved domains are C-X_2_-C-X_18–20_-C-X_2_-C. The prediction of cis-elements and the expression trend of the *PbGATA* subfamilies indicate that they participate in a series of physiological processes of different tissues, especially in coping with drought stress, which was also reflected in the subsequent qRT-PCR analysis results. *PbGATA16*, *PbGATA22*, and *PbGATA5* are vital in dealing with drought stress, whereas *PbGATA12* and *PbGATA22* are important in dealing with low-temperature stress. Fragment replication events are important in *PbGATA* gene amplification, and amplification can occur before *P. bournei*, *Arabidopsis*, and *rice* differentiation. This study provides a scientific basis for further exploration of the *GATA* family gene of *P. bournei* and also complements the comprehensive study of GATA transcription factors.

## Figures and Tables

**Figure 1 ijms-24-10342-f001:**
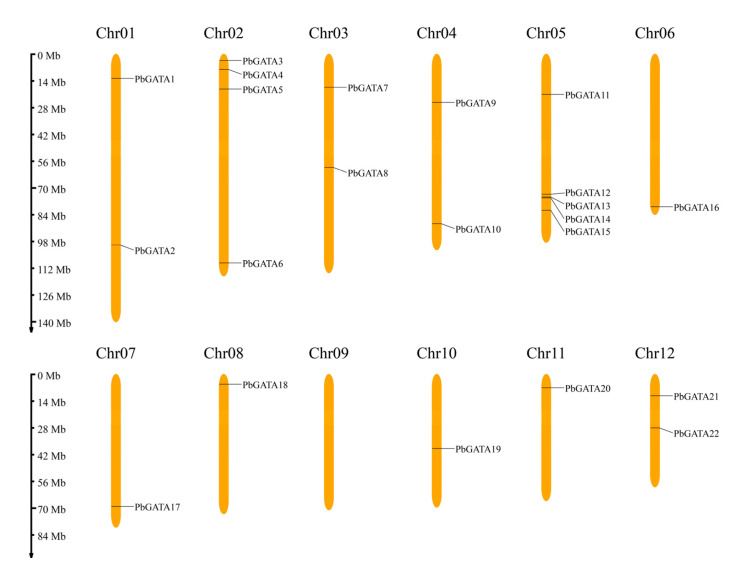
Distribution of *PbGATA* genes in *Phoebe bournei* chromosomes. Each chromosomal graphic shows the chromosome number at the top. The scale on the left can be used to assess chromosomal length and gene position.

**Figure 2 ijms-24-10342-f002:**
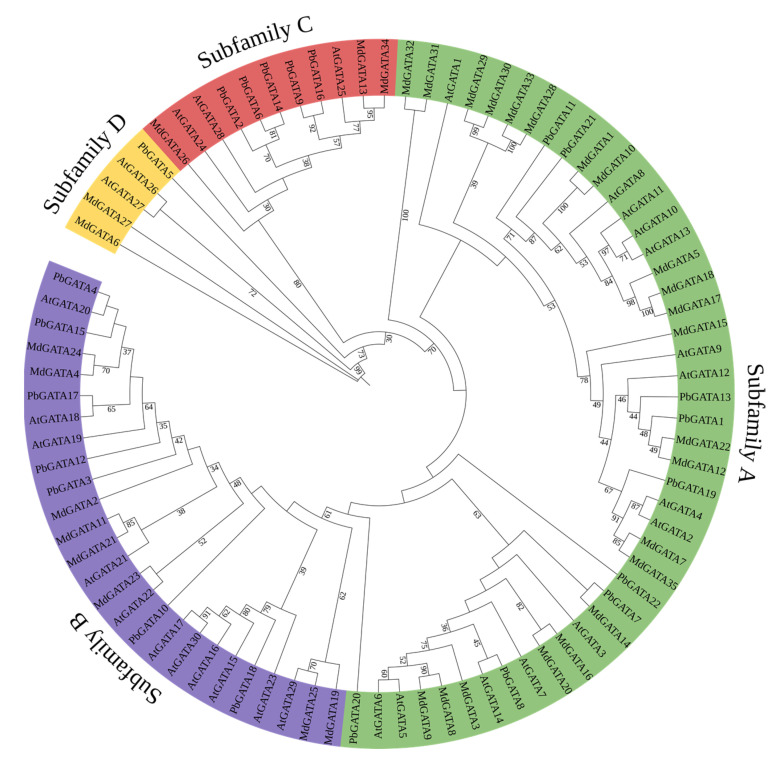
Phylogenetic tree of three plants’ GATA proteins. The different colored arcs represent the GATA protein subfamilies. The tree was built using 22 PbGATAs from *Phoebe bournei*, 30 AtGATAs from *Arabidopsis thaliana*, and 35 MdGATAs from *Malus domestica*. MEGA 7.0.21 was used to create a maximum likelihood phylogenetic tree, and the bootstrap test replicate was set to 1000 times.

**Figure 3 ijms-24-10342-f003:**
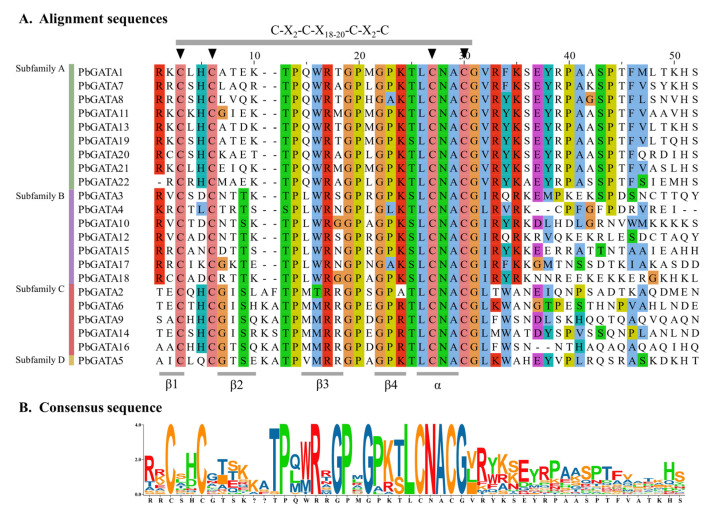
GATA domain sequence alignments of *Phoebe bournei* GATA family members. (**A**) At the top, highly conserved amino acid positions are marked with letters and triangles; (**B**) Sequence identities are shown at the bottom.

**Figure 4 ijms-24-10342-f004:**
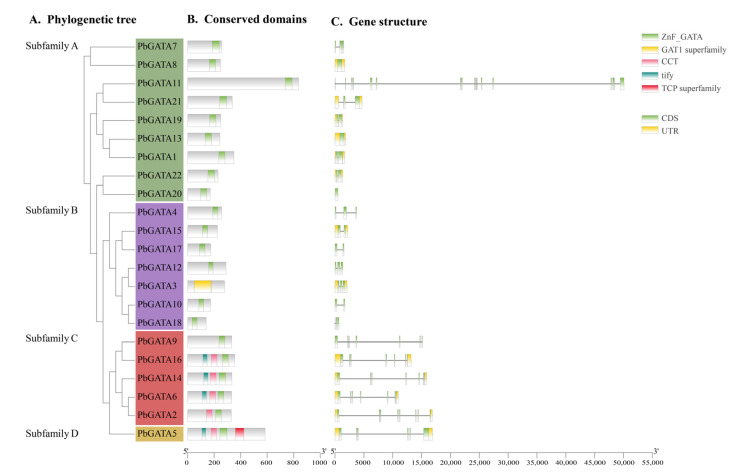
*PbGATA* gene phylogenetic relationship and gene structure schematic diagram. (**A**) Phylogenetic tree of 22 PbGATA proteins, which was constructed with MEGA 7.0.21, and the guided test replicates were set to 1000 times. (**B**) Conserved domains’ PbGATA proteins. The protein’s length can be approximated using the scale at the bottom. (**C**) Exon/intron structure of the *PbGATA* gene. The exon is represented by the green box, while the intron is represented by the black line. The yellow box indicates the *PbGATA* gene’s UTR region.

**Figure 5 ijms-24-10342-f005:**
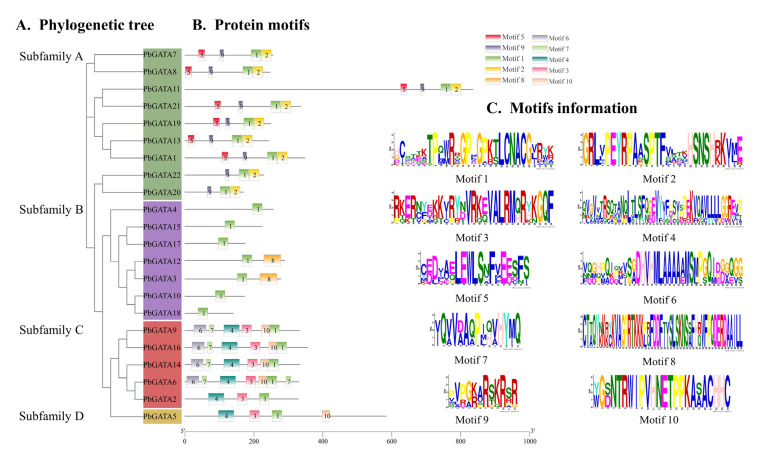
Schematic diagram of the conserved motif of the *PbGATA* gene. (**A**) Phylogenetic tree of 22 PbGATA proteins. (**B**) Different colors correspond to different types of motifs with the numbers 1–10. (**C**) The sequence information of 10 conserved motifs.

**Figure 6 ijms-24-10342-f006:**
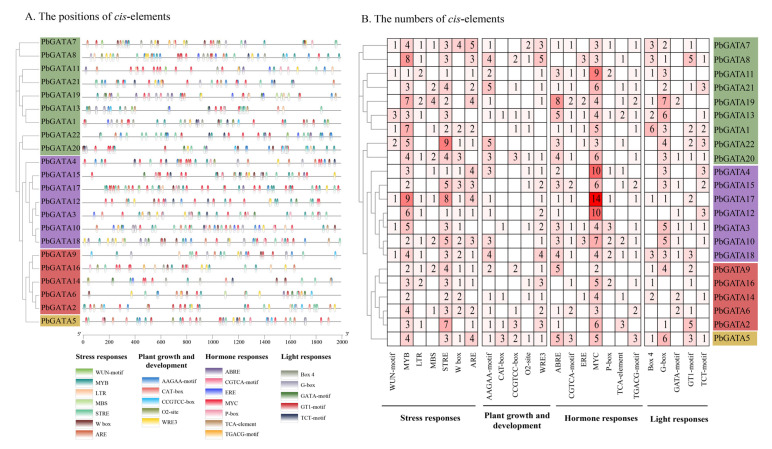
Schematic diagram of cis-element location and number. (**A**) The cis-element prediction of 22 *PbGATA* gene promoter sequences (−2000 bp) was analyzed by PlantCARE. Below are the 24 cis-elements and their classes. (**B**) The numbers of the 24 cis-elements of the 22 *PbGATA* genes.

**Figure 7 ijms-24-10342-f007:**
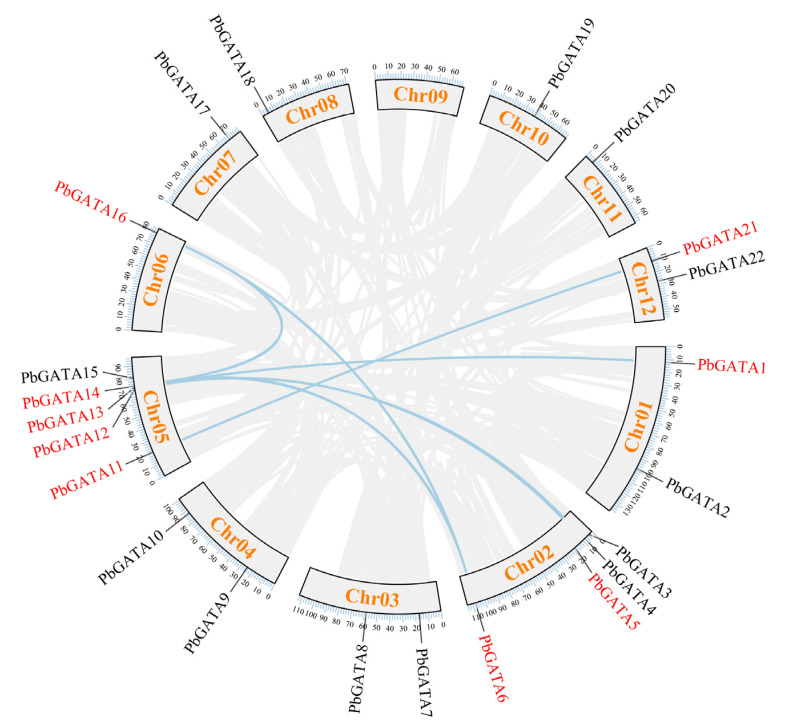
Synteny analysis of the *PbGATA* family in *Phoebe bournei*. The gray lines represent all synteny blocks in the *Phoebe bournei* genome, whereas the blue lines represent duplicated *PbGATA* gene pairs. The chromosome number is presented in a rectangular box for each chromosome.

**Figure 8 ijms-24-10342-f008:**
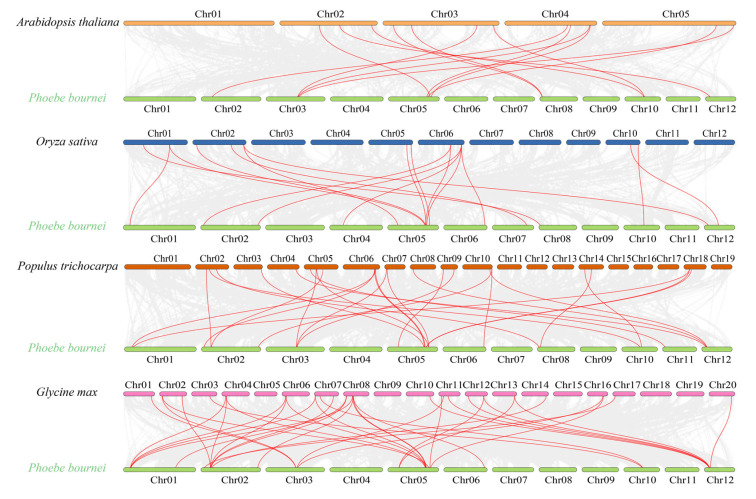
*Arabidopsis*, rice, poplar, soybean, and *Phoebe bournei* GATA gene synteny analysis. The red lines highlight the syntenic *GATA* gene pairs, while the gray lines in the background show the collinear blocks in the genomes of *Phoebe bournei* with other plants.

**Figure 9 ijms-24-10342-f009:**
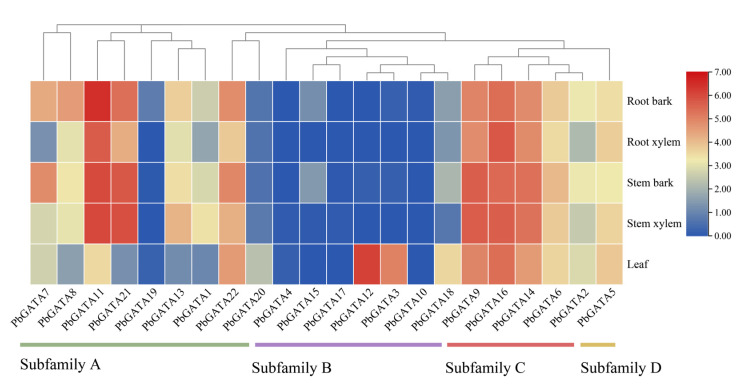
Tissue-specific gene expression patterns of 22 *PbGATA* genes, including the root bark, root xylem, stem bark, stem xylem, and leaf. The high and low transcript abundances are denoted by the red and blue colors, respectively. Expressions with log_2_ ([FPKM] + 1) normalization.

**Figure 10 ijms-24-10342-f010:**
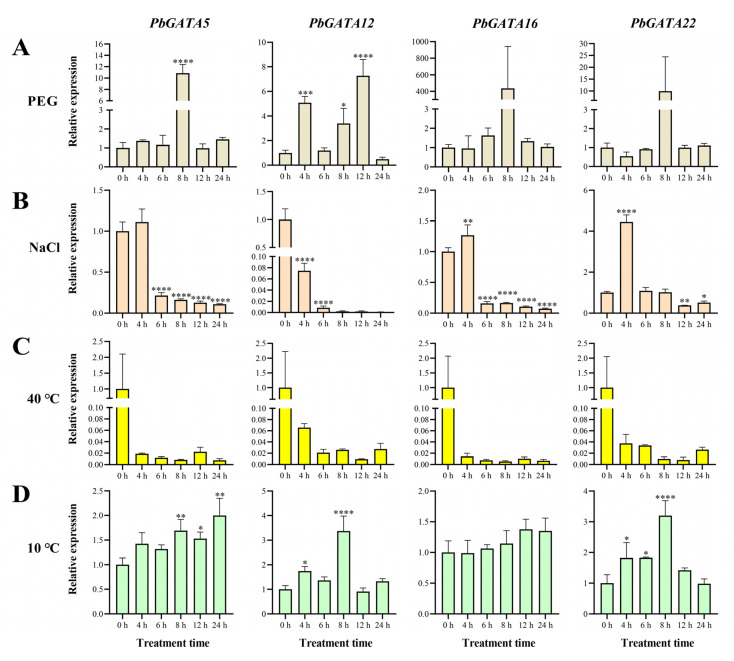
The expression profile of the *PbGATA* gene in *Phoebe bournei* was detected by qRT-PCR in response to drought, salt, and temperature stress. (**A**) Relative gene expression levels under drought (10% PEG6000) treatment over the same time period (0, 6, 8, 12, and 24 h). The control group was treated with distilled water. (**B**) Relative gene expression levels under salt (10% NaCl solution) stress. The control group was treated with distilled water. (**C**) Relative gene expression levels under heat (40 °C) stress. The control group was 25 °C, and the humidity was 75%. (**D**) Relative gene expression levels under cold (10 °C) stress. The control group was 25 °C, and the humidity was 75%. (* *p* < 0.05, ** *p* < 0.01, *** *p* < 0.0005, and **** *p* < 0.0001).

**Figure 11 ijms-24-10342-f011:**
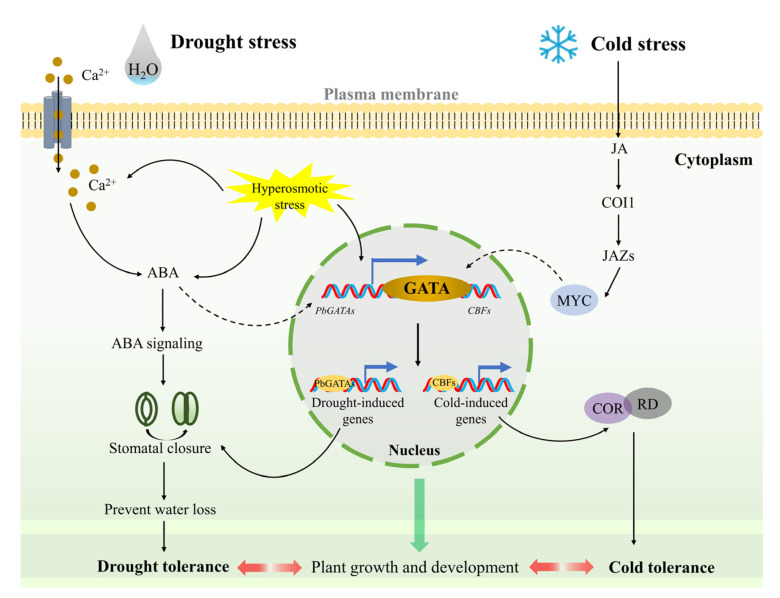
GATA signaling pathway prediction model in *Phoebe bournei* in adapting to drought and cold stress. Drought stress causes plants to create hyperosmotic stress and causes Ca^2+^ to enter cells, both of which induce ABA to accumulate and drought-related genes to express in plants, which leads to stomatal closure, allowing plants to enhance drought tolerance. Under cold stress, plants boost endogenous jasmonate synthesis, activate COI1 receptors, and target JAZ proteins for breakdown and MYC release. MYC activates downstream cold response genes via the GATA protein to increase cold tolerance.

**Table 1 ijms-24-10342-t001:** Detailed information on 22 *PbGATA* genes of *Phoebe bournei* and their encoded proteins.

Gene Accession	Gene Id	Size /aa ^1^	MW ^2^ /Da	Theoretical pI ^3^	Instability Index	Aliphatic Index	GRAVY ^4^	Subcellular Localization
OF13718-RA	*PbGATA1*	347	38,311.39	5.76	53.25	52.85	−0.728	Nuclear
OF15544-RA	*PbGATA2*	328	35,629.81	7.63	49.29	66.07	−0.498	Nuclear
OF04947-RA	*PbGATA3*	277	30,986.99	9.11	55.98	51.08	−0.758	Nuclear
OF04715-RA	*PbGATA4*	255	28,381.02	6.23	69.48	63.41	−0.765	Nuclear
OF04159-RA	*PbGATA5*	583	63,358.54	5.73	54.96	73.95	−0.540	Cytoplasmic
OF25233-RA	*PbGATA6*	330	34,979.09	6.08	51.92	65.03	−0.576	Cytoplasmic
OF12989-RA	*PbGATA7*	255	28,381.02	6.23	69.48	63.41	−0.765	Nuclear
OF24112-RA	*PbGATA8*	246	27,122.34	9.80	65.88	53.13	−0.778	Nuclear
OF29225-RA	*PbGATA9*	331	35,698.90	6.25	42.99	63.69	−0.549	Nuclear
OF20951-RA	*PbGATA10*	172	19,533.77	9.83	63.42	84.42	−0.623	Cytoplasmic
OF07140-RA	*PbGATA11*	835	94,357.56	8.70	45.27	77.02	−0.479	Chloroplast
OF10821-RA	*PbGATA12*	288	32,390.46	9.57	56.14	55.24	−0.914	Nuclear
OF26110-RA	*PbGATA13*	242	26,442.96	9.48	60.49	60.91	−0.614	Nuclear
OF26160-RA	*PbGATA14*	333	36,686.85	5.01	54.52	68.86	−0.738	Nuclear
OF09346-RA	*PbGATA15*	223	23,988.63	9.64	58.76	50.85	−0.697	Nuclear
OF29485-RA	*PbGATA16*	355	38,554.05	6.20	46.55	63.80	−0.622	Cytoplasmic
OF24473-RA	*PbGATA17*	173	18,087.43	6.51	41.87	69.31	−0.353	Cytoplasmic
OF06020-RA	*PbGATA18*	139	15,586.05	9.46	45.72	72.37	−0.732	Nuclear
OF14849-RA	*PbGATA19*	248	27,353.29	5.41	64.32	50.73	−0.706	Nuclear
OF14230-RA	*PbGATA20*	169	19,161.81	9.36	64.41	57.16	−0.938	Nuclear
OF14459-RA	*PbGATA21*	335	35,786.94	5.25	52.52	62.33	−0.527	Nuclear
OF16584-RA	*PbGATA22*	228	25,868.95	6.67	76.26	56.45	−0.885	Nuclear

^1^ aa: amino acid number; ^2^ MW: molecular weight; ^3^ pI: theoretical isoelectric point; ^4^ GRAVY: grand average of hydropathicity.

## Data Availability

All the data and materials that are required to reproduce these findings can be shared by contacting the corresponding author.

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
