# Peer review of "Genome-Wide Identification of GATA Family Genes in Phoebe bournei and Their Transcriptional Analysis under Abiotic Stresses"

_ijms, 2023, doi:10.3390/ijms241210342_

Round 1
Reviewer 1 Report
1. there is some redundancy in Introduction which should be rewritten.
2. The following sentences should be improved:
Line 60 GATA factor has the effect of inhibiting Arabidopsis flowering effect.
Line73 GNC and CGA1 regulate nitrogen assimilation, and leaves regulate plastid development and starch production
line186 It was shown that the genes of the PbGATA family played a greater role in stress and hormonal response.
3. “Research demonstrates that 268 PbGATA5, PbGATA22, and PbGATA16 have crucial functions in the response to drought 269 stress in P. bournei.”
“The expression of PbGATA12 and PbGATA22 was especially high 276 after 8 hours of treatment, indicating that PbGATA12 and PbGATA22 played an essential 277 role in P. bournei's response to low temperature stress”
A gene is responsive to stress doesn’t mean the gene plays an essential role to stress.
4. The results of Expression of all PbGATA Genes under Abiotic Stress should be analyzed and presented
5. In figure11 the authors say ABA and JA induce the expression of PbGATA genes, the authors should present result of expression of PbGATA genes to these two hormones.
Minor editing of English language required
Reviewer 2 Report
This is an interesting and comprehensive work characterizing the GATA transcription proteins in a common tree Phoebe bournei and comparing various aspects of PbGATA to those in other plants. Experiments were also performed to explore the involvement of some of the PbGATA in drought, salinity and temperature stress.
The figures are useful, but the lettering in Figs. 3, 4, 5 and 8 is too small.
All the acronyms should be explained.
The English expression is satisfactory, with occasional unusually phrased sentence. For instance line 85, 86: “Its wood aroma, tough material is not easy to crack and easy to process, widely used in wood carving art, building construction, with high economic value and ecological value.” Perhaps the authors should ask a native English speaker to go over the paper.
Reviewer 3 Report
Comments and suggestions:
1. Abstract – L.16, “found” changed to “reported” because it is possible that someone had found the GATA genes in Phoebe bournei but not reported in scientific publications. L.20, “widely distributed” changed to “clearly divided” because phylogenetic analysis is used to find relationships among different groups (clades) of studied things. Add a sentence about the chromosomal distribution of GATA genes in P. bournei. L.31, no “to” should be used after “helps” in a sentence.
2. Introduction – L.35-43. Rewrite the starting sentences, such as “The growth and development of plants is a continuous process typically starting from seed germination and ended with seed maturation. During these stages, plants must face and respond to a variety of environmental conditions. Plant responses to environmental challenges are commonly mediated through transcription factors that regulate gene expression of their target genes via cis-acting elements in the promoter. Therefore, the study of transcription factors is important to understand genetic control of gene expression in many plant metabolic pathways [1,2]. Many well known …….. (Basic Helix-Loop-Helix) and WRKY [11], ERF [12], and CBF [13].
L.45, provide the references for each after fungi, animals, and plants.
L.81, delete “production and”
Throughout the text, spell out the genus name Phoebe when it is used to start a sentence or in the legend of tables and figures.
3. Results – L.102, change “Figure 6” to “Figure 1”, and rearrange the figures accordingly.
L.271-272, …to varying degrees, but the upregulation of PbGATA16 and PbTAGA22 began at 4h after the treatment and the PbTAGA22 had higher upregulation than PbGATA16.
English writing throughout the whole manuscript needs to be checked by a native tongue expert for accuracy of word choice and grammar correction.
English writing needs careful and thorough checks for accuracy and correctness.
Reviewer 4 Report
The manuscript is written well. The data is useful for further investigation of GATA transcription factors in forest trees and other crops. The manuscript should be accepted for publication.
Author Response
Dear reviewer,
We feel great thanks for your professional review work on our manuscript titled “Genome-wide identification of GATA family genes in Phoebe bournei and their transcriptional analysis under abiotic stresses".
Thank you for affirming our article.
Yours sincerely,
Ziyuan Yin
Round 2
Reviewer 3 Report
Abstract: the chromosomal distribution should be "11 out of 12 chromosomes, except chromosome 9"
In the legends of all tables and figures, the genus names should be fully spelled out.
The English writing is fine.
